# Mediterranean Alcohol-Drinking Pattern and Arterial Hypertension in the “Seguimiento Universidad de Navarra” (SUN) Prospective Cohort Study

**DOI:** 10.3390/nu15020307

**Published:** 2023-01-07

**Authors:** Aitor Hernández-Hernández, David Oliver, Miguel Ángel Martínez-González, Miguel Ruiz-Canela, Sonia Eguaras, Estefanía Toledo, Pedro Antonio de la Rosa, Maira Bes-Rastrollo, Alfredo Gea

**Affiliations:** 1Department of Cardiology, University of Navarra Clinic, University of Navarra, 28027 Madrid, Spain; 2Department of Preventive Medicine and Public Health, School of Medicine, University of Navarra, 31008 Pamplona, Spain; 3CIBER Fisiopatología de la Obesidad y Nutrición, Instituto de Salud Carlos III, 28027 Madrid, Spain; 4IdiSNA, Instituto para la Investigación de la Salud de Navarra, 31008 Pamplona, Spain

**Keywords:** alcohol, alcohol drinking pattern, arterial hypertension, Mediterranean alcohol drinking pattern, cohort studies

## Abstract

Alcohol drinking patterns may determine the risk of hypertension and may also modify the detrimental effect of high alcohol intake. We prospectively evaluated the effect of the Mediterranean alcohol-drinking pattern and its interaction with the amount of alcohol consumed on the incidence of arterial hypertension. In the “Seguimiento Universidad de Navarra” (SUN) cohort, we followed-up 13,805 participants, all of them initially free of hypertension, during a maximum period of 16 years. Information about diet, chronic diseases, lifestyle and newly diagnosed hypertension was collected using validated questionnaires. We used a 7-item score (0 to 9 points) that jointly considered moderate alcohol consumption, distributed over the week, with meals, and a preference for red wine and avoidance of binge-drinking. During 142,404 person-years of follow-up, 1443 incident cases of hypertension were identified. Low adherence (score < 2) to the Mediterranean alcohol-drinking pattern was significantly associated with a higher incidence of hypertension (multivariable-adjusted hazard ratio 1.81, 95% confidence interval 1.09–2.99) as compared to the high-adherence (score > 7) category. Among alcohol consumers, a high adherence to the MADP is associated with a lower incidence of hypertension. Compared with abstinence, a high adherence did not seem to differ regarding its effect on hypertension risk.

## 1. Introduction

Hypertension (HT) prevalence is increasing rapidly. It is the main cardiovascular risk factor, and cardiovascular disease is the leading cause of death worldwide [1]. More than nine million deaths per year are attributed to HT complications [2]. However, some of the HT triggers are potentially modifiable. Additionally, some important modifiable risk factors such as diet or alcohol consumption are common determinants to other pathologies. Excessive alcohol consumption and a pattern characterized by binge-drinking (consumption of large quantities of alcohol in a single session) are associated with a higher incidence of cardiovascular disease, heart failure and mortality [3,4,5]. In the same way, excessive alcohol consumption is associated with an increased risk of HT in men and women [6]. Multiple mechanisms of HT development caused by excessive alcohol consumption have been described [7]. However, the relationship between light-to-moderate alcohol consumption and the incidence of HT remains controversial. A daily small amount of alcohol has been associated with a lower risk of high blood pressure in women but not in men [6,8]. Nevertheless, there are other important dimensions of alcohol consumption, such as the type of alcoholic beverage, or if it is consumed with meals, that could be relevant and whose effect over HT prevention remains uncertain [9,10]. Additionally, there is no clear evidence about which is the most effective alcohol-drinking pattern (if any) for HT prevention in healthy populations.

The Mediterranean alcohol-drinking pattern (MADP) is characterized by moderate alcohol intake, preferably red wine, with meals and without excess [11,12,13]. Alcohol consumption following the MADP has been associated with a reduction in all-cause mortality as compared to alcohol abstinence or to drinking habits showing poor adherence to this drinking pattern [14]. Similar results were found when the assessed outcomes were cardiovascular events or cardiovascular mortality [15]. These results are in contrast with those of other studies [16], pointing out the importance of considering not only the amount of alcohol consumed, but also the drinking pattern.

We aimed to prospectively assess the relationship between adherence to the MADP and the incidence of HT, as well as the role of MADP as a modifier of the effect of the amount of alcohol consumed on the incidence of HT.

## 2. Materials and Methods

### 2.1. Study Population

The SUN (Seguimiento Universidad de Navarra) project is a dynamic multipurpose cohort which started in 1999, composed of Spanish university graduates. Baseline assessment and biennial follow-up questionnaires are collected through postal or web-based questionnaires. The design and methods of this project can be found elsewhere [17]. The study protocol was approved by the Institutional Review Board of the University of Navarra.

We assessed 22,467 participants recruited before October 2015, to ensure they had completed at least the two-year follow-up questionnaire; 4399 participants were excluded because of prevalent HT (self-reported diagnosis of HT in the baseline questionnaire, systolic blood pressure (SBP) > 130 mmHg or diastolic blood pressure (DBP) > 80 mmHg or on antihypertensive treatment); 1737 participants reported total daily energy intake outside of predefined limits (<800 or >4000 Kcal/day for males, and <500 or >3500 Kcal/day for females) [18] and were excluded. Additionally, 1164 participants were excluded due to previous diagnosis of chronic diseases (cancer, diabetes or cardiovascular diseases such as myocardial infarction, stroke, revascularization procedures, valvular pathology, implanted pacemaker, or heart transplant previous to baseline). Out of the remaining 15,167 participants, 1362 were lost to follow-up (retention in the cohort: 91.0%), resulting in a final sample of 13,805 participants (Figure 1).

### 2.2. Exposure: Mediterranean Alcohol-Drinking Pattern (MADP)

Information on alcohol intake and other characteristics related to alcohol consumption were assessed with the baseline questionnaire. This questionnaire included a validated 136-item food-frequency questionnaire, with five questions about different alcoholic beverages consumption [19]. Other information about alcohol drinking habits (i.e., maximum number of drinks consumed in a single occasion) was also collected with the baseline questionnaire.

Based on this information, a 0-to-9-points score was created to capture the adherence to the traditional MADP [14]. The score accounted for seven dimensions of alcohol consumption: (a) moderate intake, (b) distribution over the week, (c) preference for wine, (d) preference for red wine, (e) consumption with meals, (f) low consumption of spirits, and (g) avoidance of binge-drinking occasions.

The score for each item was defined as follows: (a) moderate intake: 2 points if the intake was 10–50 g/day in men or 5–25 g/day in women; 1 point if the alcohol intake was below this range; 0 points if alcohol intake was beyond this range; (b) distribution over the week: we calculated the ratio between the number of drinking days per week and grams of alcohol intake during a week and categorized it in quartiles; 2 points were given to participants in the highest quartile, 1 point to participants in the second and the third quartiles, and 0 points to those in the lowest quartile; (c) preference for wine: 1 point if wine consumption accounted for at least 75% of alcohol; (d) preference for red wine: 1 point if red wine accounted for at least 75% of total wine intake; (e) consumption with meals: 1 point if 75% of wine or more is consumed with meals; (f) low consumption of spirits: 1 point if spirits conform 25% or less of total alcohol intake; (g) avoidance of binge-drinking occasions: 1 point if the maximum number of drinks consumed in a single occasion was lower than 6.

After the scores for the seven items were summed up, the total score ranged from 0 to 9 points. Participants were grouped into five categories according to their score: low adherence (0 to 1 points), low–moderate adherence (2 to 3 points), moderate adherence (4 to 5 points), moderate–high adherence (6 to 7 points) and high adherence (8 to 9 points). Abstainers were separately represented in a sixth group. As results for the low–moderate, moderate, and high–moderate categories were similar, these three categories were merged.

### 2.3. Outcome: Arterial Hypertension

In every questionnaire, participants were asked if they had been diagnosed with HT and the diagnosis date. In addition, information about systolic and diastolic blood pressure was collected at baseline, as well as data on antihypertensive treatment.

Those participants who did not have prevalent HT at baseline and reported a new medical diagnosis of HT in any of the follow-up questionnaires were considered incident cases. The self-reported diagnosis of HT had been previously validated for this cohort [20].

### 2.4. Covariates Assessment

Information about anthropometric characteristics [21], classical cardiovascular risk factors (hypercholesterolemia, diabetes), metabolic syndrome [22], lifestyle (smoking status, physical activity [23], sedentary activities), prevalent chronic diseases (cardiovascular diseases or cancer) and use of NSAID drugs [24] was collected at baseline.

Based on the food-frequency questionnaire, we also evaluated adherence to the Mediterranean dietary pattern, using a classical score without the alcohol component to avoid overlapping with the main exposure [25].

### 2.5. Statistical Analysis

We fitted Cox regression models to estimate the relationship between adherence to MADP and incidence of HT. We used the MADP high-adherence group as the reference category and estimated hazard ratios (HR) and their 95% confidence intervals (CI) for all other groups of adherences and for abstainers. In those models, the exit time was defined as the date of HT diagnosis for incident cases or date of the last follow-up questionnaire for non-cases. Age was the underlying time variable in these model.

We fitted an age- and sex-adjusted model and three multivariable models. The first multivariable model included as covariates the following potential confounders: body mass index (Kg/m^2^, quintiles), smoking habit (never, former with lower cumulative exposure, former with higher cumulative exposure, current smoker with lower cumulative exposure, current smoker with higher cumulative exposure—using median cumulative exposure to tobacco (packages-years) to classify in lower/higher cumulative exposure), leisure-time physical activity (MET-h/week), total energy intake (Kcal/d, quintiles), adherence to the Mediterranean dietary pattern (tertiles), time spent watching television (h/d, continuous), sugar-sweetened beverages consumption [26,27] (mL/d, tertiles), aspirin or NSAID treatment (yes/no). The second multivariable-adjusted model included all the covariates listed above and fried-food consumption [28] (g/d, tertiles), fast-food consumption (g/d, tertiles), and energy-adjusted sodium intake (tertiles). Additionally, the third multiple-adjusted model included energy-adjusted caffeine intake (tertiles), energy-adjusted fat-free dairy products consumption [29] (tertiles), energy-adjusted potassium intake (g/d, tertiles) and family history of hypertension (yes/no).

Moreover, we assessed the association between the MADP and the incidence of HT stratifying by categories of alcohol intake (abstainers (reference), low–moderate intake (0–<25 g/d for women, 0–50 g/d for men), high intake (>25 and >50 g/d for women and men respectively). In the same way, we used restrictive cubic splines to assess a potential non-linear association between alcohol intake and the incidence of HT in two groups of adherences to the MADP (0 to 3 points and 4 to 7 points, excluding alcohol intake from the score).

Finally, we refitted the main analysis under different scenarios to test the robustness of the results: (a) assessing only incident cases after 2 years of follow-up (left-truncating follow-up at 2 years); (b) excluding late incident cases of HT: (b1) right-truncating follow-up at 10 years; (b2) right-truncating follow-up at 8 years; (c) changing allowed limits for total energy intake: (c1) to percentiles 1st and 99th; (c2) to percentiles 5th and 95th; (d) using a less strict definition of prevalent HT (including in the study participants with baseline systolic blood pressure between 130 and 139, or diastolic blood pressure between 80 and 89 mmHg). Moreover, we conducted subgroup analysis stratifying the sample according to sex, BMI (<25/≥25 kg/m^2^), and age (<40/≥40y).

Analyses were performed using Stata version 12.0 StataCorp LP (College Station, TX, USA).

## 3. Results

A total of 1443 cases of incident HT were identified during the follow-up period (142,404 person-year; mean follow-up 10.3 years). The main characteristics of participants according to their adherence to the MADP are represented in Table 1. Participants in the highest adherence group were more likely to be older and less physically active, and less likely to smoke. Almost all variables compared were statistically different between groups.

A significant association between adherence to the MADP and the incidence of HT was observed when we compared low-adherence category (0–1) with high-adherence category (8–9) [HR (95% CI) =1.81 (1.10–2.99)] (Table 2). Point estimates for the other comparisons also pointed towards a higher risk of HT [HR (95% CI) =1.13 (0.91–1.39)], although the results were non-statistically significant.

The association between adherence to the MADP and the incidence of HT across strata of alcohol consumption as compared with abstention are shown in the Figure 2. The inverse association between adherence to the MADP and the risk of the incident HT seems to be independent from the amount of alcohol intake. However, the association of the MADP and HT was more apparent within the stratum of high alcohol intake. The increased risk of HT among drinkers with a low adherence to the MADP was stronger for participants with a high average alcohol intake.

The estimates shown in Figure 2 were adjusted for age, sex, body mass index (Kg/m^2^, quintiles), smoking habit (never, former intensive, former non-intensive, current smoker intensive, current smoker non-intensive), leisure-time physical activity (MET-h/week), total energy intake (Kcal/d, quintiles), adherence to the Mediterranean dietary pattern (tertiles), time spent watching television (h/d, continuous), sugar-sweetened beverages intake (tertiles), aspirin or NSAID treatment, fried-food consumption (tertiles), fast-food consumption (tertiles), and energy-adjusted sodium intake (tertiles). MADP score ranges from 0 to 7, as the alcohol intake item was removed.

We further deepened into the same interaction in a different way. Figure 3 shows the differential association of alcohol intake on the incidence of HT for two categories of adherence to the MADP (0–3, and 4–7). Among participants with a lower adherence to the MADP (0–3), the average amount of alcohol intake was almost linearly associated with the incidence of HT as compared with abstention. The increased risk was statistically significant beyond 25 g of alcohol consumption per day (ca. 2 drinks). The slope was steeper than the one observed for participants with a higher adherence to the MADP (4–7) (a more prudent drinking pattern). These results suggest that the association of alcohol intake may depend on the drinking pattern, however, the *p* for interaction was 0.15.

The results of sensitivity and subgroup analyses are presented in the Figure 4. Almost all analyses were consistent with the main analysis. We found a weaker and non-significant association for participants with BMI < 25 kg/m^2^ and participants younger than 40 y, probably because they are at a lower risk of HT. The association among women was more evident, but both men and women showed a lower risk of HT associated to a high adherence to the MADP. However, the interaction by sex, age or BMI were not statistically significant (*p* value for interaction: 0.31, 0.28, 0.25).

The estimates of Figure 4 were adjusted for age, sex, body mass index (Kg/m^2^, quintiles), smoking habit (never, former intensive, former non-intensive, current smoker intensive, current smoker non-intensive), leisure-time physical activity (MET-h/week), total energy intake (Kcal/d, quintiles), adherence to the Mediterranean dietary pattern (tertiles), time spent watching television (h/d, continuous), sugar-sweetened beverages intake (tertiles), aspirin or NSAID treatment, fried-food consumption (tertiles), fast-food consumption (tertiles), and energy-adjusted sodium intake (tertiles).

## 4. Discussion

Among drinkers, a high adherence to the MADP was inversely associated with the incidence of HT. Moreover, our results suggest that the MADP may modify the effect of the amount of alcohol consumed on the risk of HT, though the p value for interaction did not achieve the conventional threshold of p<0.05. The most harmful pattern was the one characterized by high alcohol intake and low adherence to the MADP. To our knowledge, this is the first study that has evaluated the effect of the overall drinking pattern on HT, considering several different dimensions of alcohol intake integrated in a combined score. Nevertheless, there are some other aspects of alcohol consumption, besides the amount of alcohol intake, that have been individually assessed in relation to high blood pressure.

Regarding the epidemiological evidence about the amount of alcohol consumed, several studies have shown that heavy alcohol intake (more than 50 g/day or ≥2 drinks/day) is associated with an increased risk of HT, while low alcohol intake (≤5 g/day or ≤1 drink/day) might be associated with a reduction in the risk among women. For men, the risk of HT seems to increase from the lowest intakes [8,30]. In our study, we found no evidence of an interaction with sex. 

Besides quantity of alcohol consumed, the frequency and distribution over the week are also important. Marques-Vidal et al. [31] found that, regardless of the quantity, a regular drinking pattern was associated with maintained values of blood pressure while an irregular drinking pattern, with excessive consumption days, was associated with blood pressure peaks that may have negative cardiovascular consequences. 

Likewise, binge-drinking, as opposed to the alcohol intake spread out over the week, was associated with a higher prevalence of HT in previous studies, most of them cross-sectional [32,33,34,35,36]. Additionally, recent studies in young population found results consistent with our findings [37,38]. A higher arterial pressure was observed among participants who consumed alcohol following a binge-drinking pattern, compared to non-binge drinkers [37]. Proposed mechanisms relate to vascular changes [39,40].

Another dimension of the proposed score is drinking with meals. Stranges et al. [9] found that drinking without meals appears to have a significant effect on HT risk independent of the amount of alcohol consumed, supporting the hypothesis proposed by Trevisan et al. [41] in a previous cross-sectional study conducted in Italy. Another recent cross-sectional study also pointed in the same direction [42]. Drinking alcohol with meals may change pharmacokinetics of alcohol (absorption and elimination), leading to a lower blood alcohol concentration and a lower peak [43,44]. Moreover, the interaction between alcohol and food intake showed potential benefits in small feeding trials [45,46].

Regarding the potential association between the type of alcoholic beverage and HT, there is no clear evidence to support the consumption of any specific beverage compared to the others [31,47]. However, wine consumption has been shown to reduce mortality in both normotensive and hypertensive participants [48]. Gepner et al. reported no differences in blood pressure after a 6-month intervention with 150 mL of red wine vs. water [49]. Wine intake was associated with a lower risk of HT compared with other alcoholic beverages in a previous report from our cohort [50]. Red wine polyphenol extracts may explain the beneficial impact on HT [51,52], compared to other beverages. It has been suggested that moderate wine drinkers tend to exhibit other healthy habits (higher consumption of fruits and vegetables, more physical activity, less likelihood of smoking) that could be considered possible confounders of the alcohol effect [53]. Nevertheless, we controlled for a wide range of lifestyle variables in the multivariable analysis to avoid this potential confounding.

In previous studies, which only considered the amount of alcohol consumed, women with low alcohol intake showed a lower risk of HT, but this association was not found in men. However, in our study, we found consistent results across sex strata and no significant interaction by sex. On the other hand, we found a more apparent association among older participants and participants with higher BMI. We believe that younger participants and those with lower BMI could also benefit from changing towards a healthier drinking pattern, but the association in our study was not statistically significant, probably due to a lack of power—lower number of events—in these subgroups.

According to our results, moderately drinking alcohol while maintaining the features of a high adherence to the MADP was associated with a lower risk of HT compared with departures from this drinking pattern. However, another aspect that deserves attention is alcohol abstinence. The comparison with abstainers is controversial due to some special characteristics of this group and the possibility of being former drinkers [54]. In our study, we used the participants with a high adherence to the MADP as reference category and therefore, the potential bias would only affect to the estimation corresponding to the abstainers group. Thus, we did not observe significant differences between abstainers and drinkers who adhered to the MADP regarding HT risk. More studies are needed to support this null association.

Our study shows some limitations that should be noted. First, our cohort does not constitute a representative sample of the general population, and hence, generalization of the results must be based on biological mechanism instead of statistical representativeness. Second, some degree of misclassification is possible because both exposure and outcome information were self-reported. In studies with self-reported alcohol consumption, there is room for under-reporting [55]. Nevertheless, if there was some degree of under-reporting, we would expect it to lead to a non-differential misclassification and hence the findings would most likely be biased towards a null association. The information regarding alcohol intake was collected with a previously validated food-frequency questionnaire [19] and the correlation coefficient for alcohol intake versus repeated food records was high (r = 0.89) [56]. Moreover, the main variables, including HT, have been previously validated for this cohort [20]. On the other hand, the high educational level of participants allows us to assume a better understanding and a higher accuracy in the self-reported information in the questionnaire. Participants who were lost to follow-up had a slightly lower adherence to the MADP, however there was no significant difference in total alcohol intake nor in the percentage of abstainers between participants followed and lost to follow-up. Even though the size of the cohort is large, some study groups (especially those with lower adherence to the MADP) are relatively small. The SUN cohort is formed mostly of young participants; therefore, the incidence of hypertension in the study is low. Finally, residual confounding is still possible although we adjusted for multiple potential confounders.

On the other hand, our study shows several important strengths. The sample size is large and the follow-up is long. Additionally, the retention in the cohort is 91.0%. Participants in this cohort are volunteers and highly educated subjects and more than half of them are health professionals. These facts reduce the potential for confounding by educational levels and lead to a better quality of the self-reported data, improving the internal validity of the study. Finally, the baseline questionnaire collects information about several alcohol consumption dimensions that has allowed us to evaluate the alcohol intake from a multidimensional perspective.

Drinking alcohol involves more dimensions besides the amount of alcohol consumed. Therefore, probably the most correct and complete way to assess the association between alcohol-drinking choices and health is to consider the alcohol consumption integrated into a dietary pattern. This approach takes into account possible synergies and interactions, trying to avoid confounding by other aspects of the drinking pattern. Additionally, it provides a more realistic panorama of the consumption among the population and provides a more practical basis to elaborate recommendations [57].

## 5. Conclusions

Alcohol consumed following a high adherence to the MADP is associated with a lower incidence of HT as compared to low adherence to this pattern. In the same way, the increased risk of HT associated to high alcohol consumption is higher if the adherence to the MADP is low. Compared with abstinence, alcohol consumption with a low adherence to the MADP showed a significant increase in the risk of HT, while moderate alcohol intake with a high adherence to the pattern did not seem to differ from abstinence, although more studies are needed to confirm these results.

## Figures and Tables

**Figure 1 nutrients-15-00307-f001:**
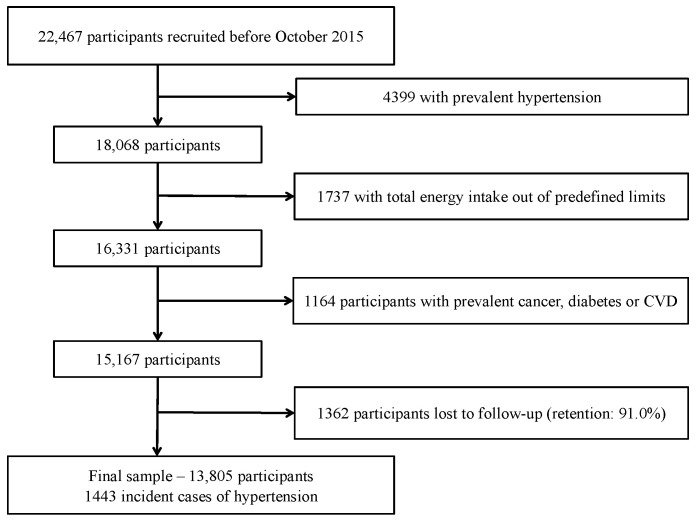
Flow-chart of participants. The “Seguimiento Universidad de Navarra” (SUN) cohort (1999–2018).

**Figure 2 nutrients-15-00307-f002:**
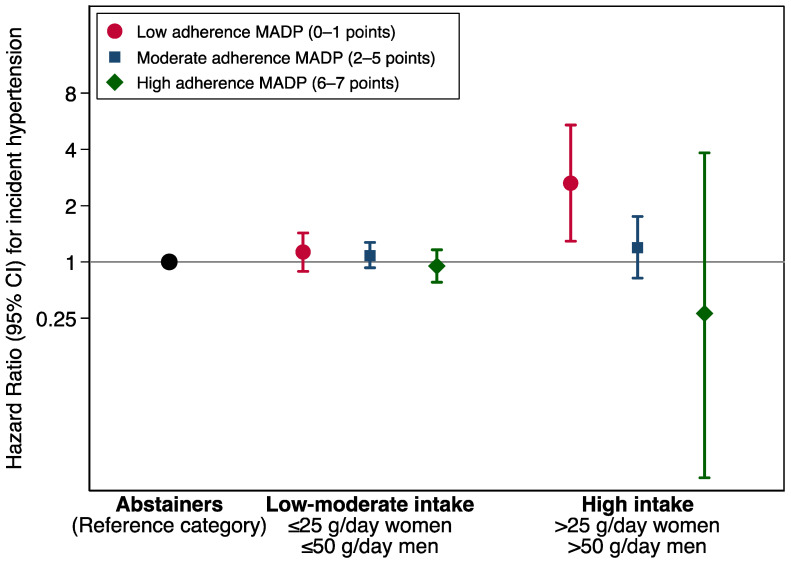
Hazard ratios (95% CI) for hypertension incidence according to Mediterranean alcohol drinking pattern adherence categories, stratified by alcohol intake. Alcohol intake item was removed from the calculation as the analysis was stratified for alcohol intake. The SUN Project 1999–2018.

**Figure 3 nutrients-15-00307-f003:**
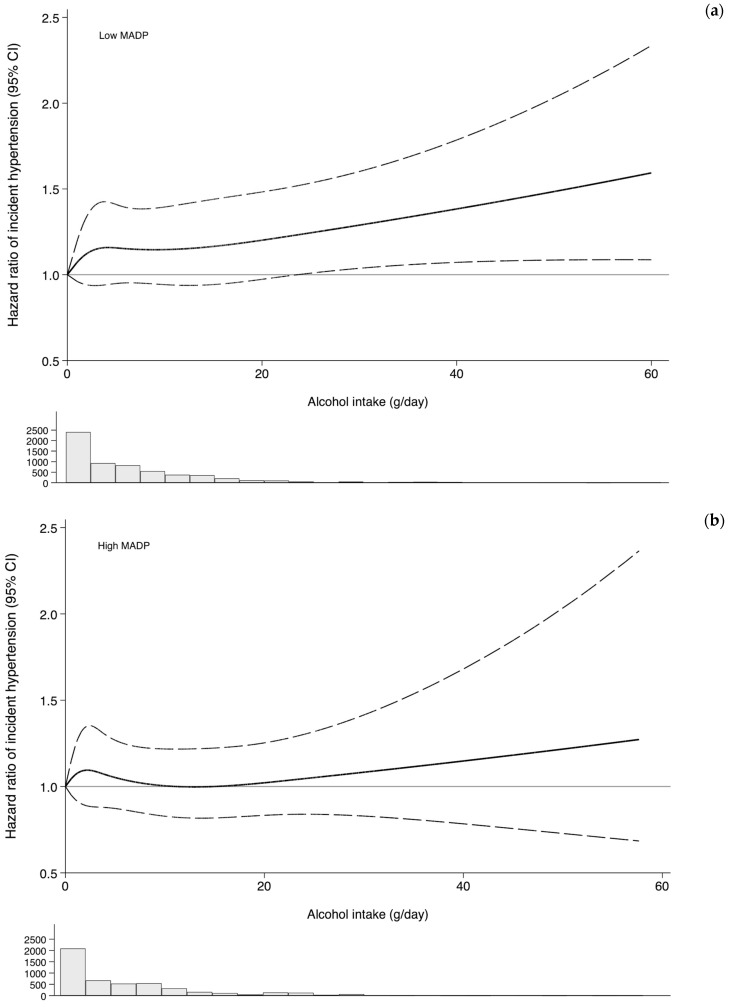
Hazard ratios (solid line) and 95% confidence interval (dashed lines) of incidence hypertension according to alcohol intake. Dose-response relationship, stratified by Mediterranean alcohol drinking pattern adherence ((**a**) low MADP = 0–3; (**b**) high MADP = 4–7) using restricted cubic splines. The histogram below the graph represents the distribution of participants by alcohol intake levels. The SUN Project 1999–2018. Adjusted for age, sex, body mass index (Kg/m^2^, quintiles), smoking habit (never, former intensive, former non-intensive, current smoker intensive, current smoker non-intensive), leisure-time physical activity (MET-h/week), total energy intake (Kcal/d, quintiles), adherence to the Mediterranean dietary pattern (tertiles), time spent watching television (h/d, continuous), sugar-sweetened beverages intake (tertiles), aspirin or NSAID treatment, fried-food consumption (tertiles), fast-food consumption (tertiles), and energy-adjusted sodium intake (tertiles).

**Figure 4 nutrients-15-00307-f004:**
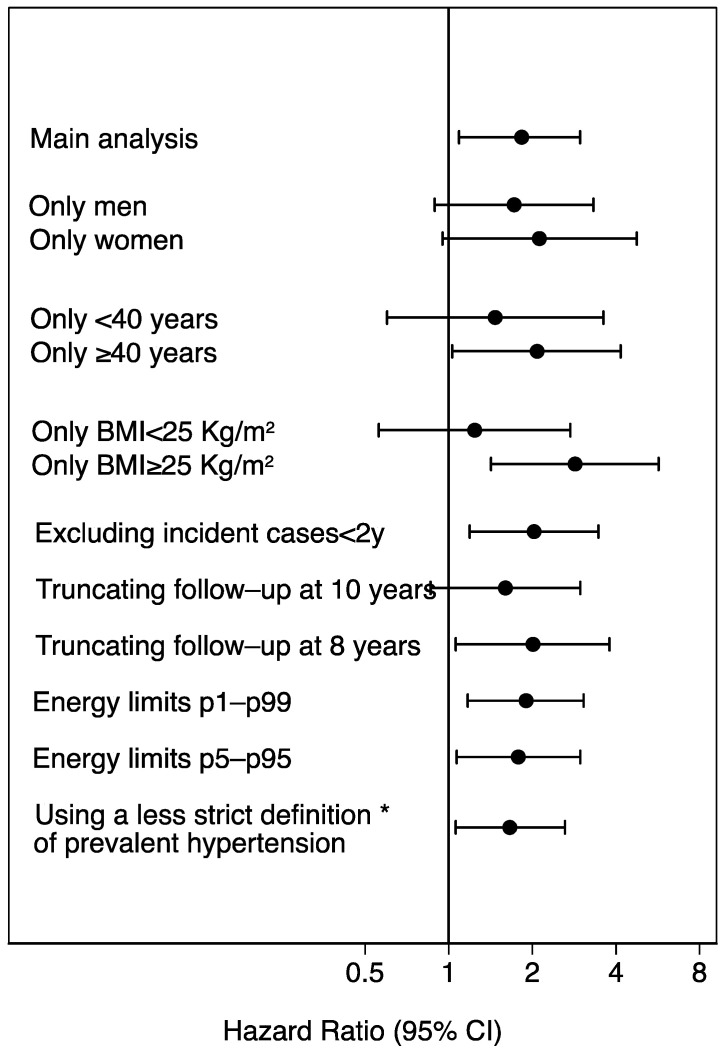
Sensitivity analyses. Hazard ratios (95% confidence interval) of incident hypertension for low (0–1) vs. high adherence (8–9) to the Mediterranean alcohol drinking pattern. The SUN Project 1999–2018. * Including also participants with baseline systolic blood pressure between 130 and 139, or diastolic blood pressure between 80 and 89 mmHg. BMI: body mass index.

**Table 1 nutrients-15-00307-t001:** Baseline characteristics of participants according to their adherence to the Mediterranean alcohol-drinking pattern score. The “Seguimiento Universidad de Navarra” (SUN) cohort 1999–2018.

		Mediterranean Alcohol-Drinking Pattern Score Adherence			
	Abstainers	Low (0–1)	Moderate (2–7)	High (8–9)	*p*-Value
N	2597	156	10,341	711	-
Age (years)	34 (10)	28 (8)	35 (11)	42 (10)	<0.0001
Sex (Female %)	84.4	43.0	63.5	65.0	<0.0001
Alcohol intake (g/d)	-	14.8 (22.5)	7.9 (9.5)	8.3 (6.6)	<0.0001
Wine (% alcohol from wine)	-	12.4 (17.2)	38.5 (34.1)	92.1(10.6)	<0.0001
Red wine (% red wine of total wine)	-	23.2 (31.8)	51.1 (43.4)	85.9 (31.3)	<0.0001
Beer (% alcohol from beer)	-	39.0 (24.0)	40.0 (33.7)	6.5 (10.0)	<0.0001
Spirits (% alcohol from spirits)	-	48.6 (21.3)	21.5 (29.5)	1.4 (4.0)	<0.0001
Ever had >5 drinks in one occasion (%)	-	97.4	36.5	2.3	<0.0001
Former smokers (%)	14.9	13.7	22.6	28.3	<0.0001
Low-grade * former smoker (%)	6.0	5.9	8.3	12.2	
High-grade * former smoker (%)	8.9	7.8	14.3	16.1	
Current smokers (%)	17.1	41.2	30.3	20.2	
Low-grade * current smoker (%)	7.4	24.2	15.7	6.0	
High-grade * current smoker (%)	9.6	17.0	14.6	14.2	
Body-mass index (kg/m^2^)	22.4 (3.1)	23.7 (3.3)	23.0 (3.2)	23.2 (2.9)	<0.0001
Prevalent Hypercholesterolemia (%)	11.9	7.1	13.0	18.6	<0.0001
Prevalent Hypertriglyceridemia (%)	3.2	3.2	4.5	3.7	0.016
Family history of Hypertension (%)	39.7	34.0	38.5	40.1	0.362
Leisure-time physical activity (METs-h/week)	14.4 (6.8–27.9)	20.0 (8.7–35.3)	17.2 (8.2–30.8)	14.9 (7.8–27.7)	<0.0001
Time spent watching television (h/d)	1.6 (1.3)	1.9 (1.2)	1.6 (1.2)	1.3 (1.1)	<0.0001
Total energy intake (Kcal/d)	2294 (617)	2489 (645)	2364 (604)	2336 (589)	<0.0001
Weight gain > 3 kg in previous year (%)	24.7	37.2	29.7	27.6	<0.0001
Mediterranean dietary pattern adherence (0–8)	3.8 (1.7)	3.8 (1.7)	4.0 (1.7)	4.0 (1.7)	0.0002
Fast-food consumption (g/d)	17 (7–29)	23 (17–41)	20 (10–32)	17 (3–29)	<0.0001
Fried-food consumption (servings/week)	3 (1–4)	3 (1–4)	3 (1–4)	3.5 (1.5–6)	0.019
Sodium intake (g/d)	3.4 (2.5–4.4)	3.7 (2.8–4.5)	3.5 (2.7–4.6)	3.4 (2.6–4.5)	0.0004
Potassium intake (g/d)	4.8 (1.6)	4.6 (1.6)	4.7 (1.5)	5.0 (1.8)	0.0009
Fruit intake (g/d)	376 (314)	287 (292)	338 (283)	388 (323)	<0.0001
Vegetable intake (g/d)	515 (343)	491 (335)	524 (324)	581 (429)	<0.0001
Fiber intake (g/d)	28.2 (12.5)	25.7 (12.5)	27.4 (11.8)	30.0 (14.3)	<0.0001
Sugar-sweetened beverages consumption (mL/d)	13 (0–86)	86 (29–157)	29 (13–86)	13 (0–29)	<0.0001
Low-fat dairy products intake (g/d)	200 (7.1–500)	69.6 (3.3–217.3)	200 (11.7–325)	101 (3.3–275)	<0.0001
NSAID treatment (%)	9.4	8.3	10.8	10.6	0.169
Caffeine intake (mg/d)	25 (4–55)	39 (18–75)	32 (14–73)	25 (4–55)	<0.0001

Numbers are means (sd), median (interquartile range) or frequencies. * Low-/high-grade smokers and former smokers are defined based on the median cumulative number of cigarettes smoked. MET: metabolic equivalents, NSAID: Non-steroidal anti-inflammatory drugs.

**Table 2 nutrients-15-00307-t002:** Association between adherence to the MADP and HT incidence.

		Mediterranean Alcohol-Drinking Pattern Adherence		
	Abstainers	Low (0–1)	Moderate (2–7)	High (8–9)
Cases/person-years	210/26,922	19/1554	1115/106,258	99/7670
Age- and sex-adjusted model	1.08 (0.84–1.37)	2.08 (1.26–3.42)	1.17 (0.95–1.44)	1 (Ref.)
Multivariable-adjusted model ^1^	1.05 (0.82–1.34)	1.83 (1.11–3.02)	1.12 (0.91–1.39)	1 (Ref.)
Multivariable-adjusted model ^2^	1.04 (0.82–1.33)	1.81 (1.10–2.99)	1.13 (0.91–1.39)	1 (Ref.)
Multivariable-adjusted model ^3^	1.04 (0.81–1.33)	1.81 (1.09–2.99)	1.13 (0.91–1.39)	1 (Ref.)

^1^. Adjusted for age, sex, body mass index (Kg/m^2^, quintiles), smoking habit (never, former intensive, former non-intensive, current smoker intensive, current smoker non-intensive), leisure-time physical activity (MET-h/week), total energy intake (Kcal/d, quintiles), adherence to the Mediterranean dietary pattern (tertiles), time spent watching television (h/d, continuous), sugar-sweetened beverages intake (tertiles), aspirin or NSAID treatment. ^2^. Additionally adjusted for fried-food consumption (tertiles), fast-food consumption (tertiles), and energy-adjusted sodium intake (tertiles). ^3^. Additionally adjusted for caffeine intake (tertiles), energy-adjusted fat free dairy products consumption (tertiles), energy-adjusted potassium intake (tertiles) and family history of hypertension (yes/no).

## Data Availability

The data presented in this study are available on request from the corresponding authors Maira Bes-Rastrollo (mbes@unav.es) and Alfredo Gea (ageas@unav.es).

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
