# Peer review of "Mediterranean Alcohol-Drinking Pattern and Arterial Hypertension in the “Seguimiento Universidad de Navarra” (SUN) Prospective Cohort Study"

_nutrients, 2023, doi:10.3390/nu15020307_

Round 1
Reviewer 1 Report
The study is novel and the results are potentially important.
However, I have some remarks:
Abstract: The authors are using the concept „…seven dimensions of alcohol consumption”. It is clarified in the text but not in the abstract. In the abstract only low/high adherence is mentioned. I suggest either removing that concept from the abstract or clarify it and include the results.
I would also suggest clarifying shortly in the abstract what „Mediterranean alcohol-drinking pattern” is and what it scores. Many readers may be not familiar to that concept developed previously by the authors.
Introduction: In some statements authors claim that moderate (?) alcohol consumption is related to reduction of all-cause mortality („Alcohol consumption following the MADP has been associated with a reduction in all-cause mortality as compared to alcohol abstinence…” or „…wine consumption has been shown to reduce mortality …”). I would be more careful in making such statements because as some studies (that included Mediterranean population) have shown that every amount of consumed alcohol is related to increased all-cause mortality (GBD 2016 Alcohol Collaborators. Alcohol use and burden for 195 countries and territories, 1990-2016: a systematic analysis for the Global Burden of Disease Study 2016. Lancet. 2018 Sep 22;392(10152):1015-1035. doi: 10.1016/S0140-6736(18)31310-2. )
Methods:
high/low grade smokers is a rarely used term. I would replace it with exposure pack-years
Analgesic treatment (almost 10% of its)- what you you precisely mean? NSAID’s? Aspirin? Opiates? How long?
Results:
Some data seem not to have normal distribution (eg. Leisure-time physical activity , Fried-food consumption). Did you check you check for that? In such cases I would you means and IQR’s
Table 1: the authors claim that high adherence patients were „…MORE LIKELY to be older and less physically active, and less likely to smoke” was it significant? The significance of differences is not mentioned in the table.
The statement: „Point estimates for the other comparisons also pointed towards a higher risk of HT [HR (95% CI) =1.13 (0.91-1.39)], although the confidence interval precludes any firm conclusion.” is an overstatement. The CI’s indicate no significance, therefore no difference in the risk.
The limitations of the study: it should be mentioned that 9% (1362 participants) were lost to follow-up. Is it possible that low-adherent to MADP patients that were high-drinkers of alcohol were lost to follow-up, but suffered from harmful effects of alcohol (including hypertension)?
Author Response
The study is novel and the results are potentially important.
However, I have some remarks:
Abstract: The authors are using the concept „…seven dimensions of alcohol consumption”. It is clarified in the text but not in the abstract. In the abstract only low/high adherence is mentioned. I suggest either removing that concept from the abstract or clarify it and include the results.
I would also suggest clarifying shortly in the abstract what „Mediterranean alcohol-drinking pattern” is and what it scores. Many readers may be not familiar to that concept developed previously by the authors.
Thank you for your comments. We modified the abstract to include this information.
Introduction: In some statements authors claim that moderate (?) alcohol consumption is related to reduction of all-cause mortality („Alcohol consumption following the MADP has been associated with a reduction in all-cause mortality as compared to alcohol abstinence…” or „…wine consumption has been shown to reduce mortality …”). I would be more careful in making such statements because as some studies (that included Mediterranean population) have shown that every amount of consumed alcohol is related to increased all-cause mortality (GBD 2016 Alcohol Collaborators. Alcohol use and burden for 195 countries and territories, 1990-2016: a systematic analysis for the Global Burden of Disease Study 2016. Lancet. 2018 Sep 22;392(10152):1015-1035. doi: 10.1016/S0140-6736(18)31310-2. )
Thank you for the suggestion. We modified the introduction accordingly.
Methods:
high/low grade smokers is a rarely used term. I would replace it with exposure pack-years
Analgesic treatment (almost 10% of its)- what you you precisely mean? NSAID’s? Aspirin? Opiates? How long?
Thank you. We further explain both covariates in the corresponding section of the new version of the manuscript.
Results:
Some data seem not to have normal distribution (eg. Leisure-time physical activity , Fried-food consumption). Did you check you check for that? In such cases I would you means and IQR’s
We updated data in table 1 to median and IQR for the quantitative variables that separated from the normal distribution.
Table 1: the authors claim that high adherence patients were „…MORE LIKELY to be older and less physically active, and less likely to smoke” was it significant? The significance of differences is not mentioned in the table.
P-values have been included for all variables in table 1, according to your suggestion.
The statement: „Point estimates for the other comparisons also pointed towards a higher risk of HT [HR (95% CI) =1.13 (0.91-1.39)], although the confidence interval precludes any firm conclusion.” is an overstatement. The CI’s indicate no significance, therefore no difference in the risk.
Thank you for the suggestion. We rephrased this sentence.
The limitations of the study: it should be mentioned that 9% (1362 participants) were lost to follow-up. Is it possible that low-adherent to MADP patients that were high-drinkers of alcohol were lost to follow-up, but suffered from harmful effects of alcohol (including hypertension)?
Participants that were lost to follow-up had a slightly lower adherence to the MADP, but there was no statistically significant difference in mean alcohol intake nor in the percentage of abstainers. Nevertheless, we included this information in the limitation section of the new version of the manuscript.
Reviewer 2 Report
This cohort study is the first to investigate the association between drinking patterns and risk of hypertension. This is a very relevant topic, considering that hypertension is and will continue to be a substantial problem in medical practice. The manuscript has a concise narrative and an overall solid methodology. However, there are some issues that need to be addressed:
· Page 3, lines 94–95: In item (a) of the MADP Score, alcohol intake below 10–50 g/d in men and 5–25 g/d in women is given 1 point. I think an explanation is required why this consumption range is scored as less adherent to MADP.
· Page 3, line 102: Please clarify why only consumption of wine is considered for item (e).
· Table 1: Please add a row showing the results of group comparisons for each characteristic with p-values. For instance, the high adherence group seems to contain more female participants than the low adherence group. Alternatively, if the findings reported in lines 168–170 are the only differences between the study groups, please add a respective statement.
· Figure 2: For this figure, a modified MADP Score is used (as explained on page 7, lines 208–209). This change obviously leads to the groups in the figure being others than those shown in Table 2. For instance, a person with high alcohol intake (score of 0 in the first MADP item) can at best score 7 in the total score, but now there are participants with high intake and high adherence. This should be stated in the manuscript. Also, to avoid confusion, I suggest adding a note about the changed scoring directly in the figure / figure caption.
· Figure 4: The manuscript states that both sensitivity and subgroup analyses are shown in Figure 4 (stated on page 7, line 230). The figure itself seems to only show sensitivity analyses, as differences between subgroups (sex, age, BMI) can only be guessed. The authors report the result for the interaction by sex in the manuscript (page 7, lines 235–236) but do not report the results for the other subgroup analyses.
· Page 9, line 285: There seems to be an issue with the citation formatting.
· Page 10, lines 328–329: The authors state that the large sample size is a strength of the study. While the total sample is indeed large, the most important conclusions are actually based on a small set of cases (19 and 99 hypertensives in the low and high adherence groups, respectively). I think this should be acknowledged in the discussion.
Author Response
This cohort study is the first to investigate the association between drinking patterns and risk of hypertension. This is a very relevant topic, considering that hypertension is and will continue to be a substantial problem in medical practice. The manuscript has a concise narrative and an overall solid methodology. However, there are some issues that need to be addressed:
- Page 3, lines 94–95: In item (a) of the MADP Score, alcohol intake below 10–50 g/d in men and 5–25 g/d in women is given 1 point. I think an explanation is required why this consumption range is scored as less adherent to MADP.
- Page 3, line 102: Please clarify why only consumption of wine is considered for item (e).
Thank you for your suggestions. We used an already defined score that capture adherence to the traditional Mediterranean way of drinking. The cut-off points were defined based on literature (Trichopoulou A, Costacou T, Bamia C, Trichopoulos D. Adherence to a Mediterranean diet and survival in a Greek population. N Engl J Med. 2003;348(26):2599-608.). Both a consumption above or below these limits would depart from the Mediterranean way of drinking. A lower alcohol intake may still be similar to the drinking pattern that the score represents, so 1 point is given to those participants in that category. However, alcohol intake over those limits clearly departs from the referred drinking pattern and therefore 0 points are given to the participants in that category.
Similarly, the score only considered wine because the score aims to capture the Mediterranean way of drinking.
- Table 1: Please add a row showing the results of group comparisons for each characteristic with p-values. For instance, the high adherence group seems to contain more female participants than the low adherence group. Alternatively, if the findings reported in lines 168–170 are the only differences between the study groups, please add a respective statement.
Thank you for the suggestion. We included p-values in table 1. Almost all the comparisons were statistically significant. We included this information in the new version of the manuscript.
- Figure 2: For this figure, a modified MADP Score is used (as explained on page 7, lines 208–209). This change obviously leads to the groups in the figure being others than those shown in Table 2. For instance, a person with high alcohol intake (score of 0 in the first MADP item) can at best score 7 in the total score, but now there are participants with high intake and high adherence. This should be stated in the manuscript. Also, to avoid confusion, I suggest adding a note about the changed scoring directly in the figure / figure caption.
Thank you for your suggestions. The information is now included in the new version of the manuscript.
- Figure 4: The manuscript states that both sensitivity and subgroup analyses are shown in Figure 4 (stated on page 7, line 230). The figure itself seems to only show sensitivity analyses, as differences between subgroups (sex, age, BMI) can only be guessed. The authors report the result for the interaction by sex in the manuscript (page 7, lines 235–236) but do not report the results for the other subgroup analyses.
The p-values for the interaction for all subgroup analyses have now been added.
- Page 9, line 285: There seems to be an issue with the citation formatting.
Thank you for noticing. It has been corrected.
- Page 10, lines 328–329: The authors state that the large sample size is a strength of the study. While the total sample is indeed large, the most important conclusions are actually based on a small set of cases (19 and 99 hypertensives in the low and high adherence groups, respectively). I think this should be acknowledged in the discussion.
We included in the limitation section this thoughts, according to your suggestion.
Reviewer 3 Report
The aim of this manuscript was to evaluate the effect of a Mediterranean alcohol-drinking pattern on the development of incident hypertension from the prospective SUN cohort study. The authors took a unique approach by calculating a score that reflects adherence to a traditional Mediterranean Alcohol-Drinking Pattern (MADP) and then examined incident hypertension (HTN) (defined as self-reported diagnosis of HTN or SBP >130/ DBP > 80 mm Hg). The statistical methods seem to be rigorous and importantly, numerous confounding variables were controlled for in the analysis. Minor comments are noted below.
11. Reference #5 is very old – I would another more recent article or meta-analysis.
22. Occasionally the authors note ‘drinking pattern’ (eg., line 197 “However, the association of the drinking pattern was more apparent in the strata of high alcohol intake.”). I think they mean “the association of the MADP was more apparent in the strata of high alcohol intake.” The authors can not make any specific comments about alcohol drinking patterns per se, such as binge drinking, but when then use the terms ‘drinking pattern’ it implies this. The drinking pattern in this study is really ‘adherence to the Mediterranean alcohol-drinking pattern’.
33. Figure 3 – the y axis font is hard to read and it is unclear what the small bar graph underneath the Hazard ratios is indicating (I would recommend deleting this graph).
4. Carefully review the manuscript for verb tense and 'plural' (an occasional 's' left off, line 89 dimension should be 'dimensions'.
Author Response
The aim of this manuscript was to evaluate the effect of a Mediterranean alcohol-drinking pattern on the development of incident hypertension from the prospective SUN cohort study. The authors took a unique approach by calculating a score that reflects adherence to a traditional Mediterranean Alcohol-Drinking Pattern (MADP) and then examined incident hypertension (HTN) (defined as self-reported diagnosis of HTN or SBP >130/ DBP > 80 mm Hg). The statistical methods seem to be rigorous and importantly, numerous confounding variables were controlled for in the analysis. Minor comments are noted below.
- Reference #5 is very old – I would another more recent article or meta-analysis.
Thank you for your suggestion. We have changed this reference to a more recente one.
- Occasionally the authors note ‘drinking pattern’ (eg., line 197 “However, the association of the drinking pattern was more apparent in the strata of high alcohol intake.”). I think they mean “the association of the MADP was more apparent in the strata of high alcohol intake.” The authors can not make any specific comments about alcohol drinking patterns per se, such as binge drinking, but when then use the terms ‘drinking pattern’ it implies this. The drinking pattern in this study is really ‘adherence to the Mediterranean alcohol-drinking pattern’.
We updated the manuscript according to your suggestion.
- Figure 3 – the y axis font is hard to read and it is unclear what the small bar graph underneath the Hazard ratios is indicating (I would recommend deleting this graph).
The figures are now bigger in the new version of the manuscript so it is easier to read.
- Carefully review the manuscript for verb tense and 'plural' (an occasional 's' left off, line 89 dimension should be 'dimensions'.
The figures are now bigger in the new version of the manuscript so it is easier to read.
Round 2
Reviewer 1 Report
The manuscript was improved and should be published.